# The Voltage-Gated Hv1 H^+^ Channel Is Expressed in Tumor-Infiltrating Myeloid-Derived Suppressor Cells

**DOI:** 10.3390/ijms24076216

**Published:** 2023-03-25

**Authors:** Marco Cozzolino, Adrienn Gyöngyösi, Eva Korpos, Peter Gogolak, Muhammad Umair Naseem, Judit Kállai, Arpad Lanyi, Gyorgy Panyi

**Affiliations:** 1Department of Biophysics and Cell Biology, Faculty of Medicine, University of Debrecen, 4032 Debrecen, Hungary; marco.cozzolino@med.unideb.hu (M.C.); korpos.eva@med.unideb.hu (E.K.); umair.naseem@med.unideb.hu (M.U.N.); 2Department of Immunology, Faculty of Medicine, University of Debrecen, 4032 Debrecen, Hungary; gadrienn@med.unideb.hu (A.G.); gogy@med.unideb.hu (P.G.); immunology.office@med.unideb.hu (J.K.); alanyi@med.unideb.hu (A.L.); 3ELKH-DE Cell Biology and Signaling Research Group, Faculty of Medicine, University of Debrecen, 4032 Debrecen, Hungary

**Keywords:** Myeloid-derived suppressor cell, Hv1 proton channel, tumor microenvironment

## Abstract

Myeloid-derived suppressor cells (MDSCs) are key determinants of the immunosuppressive microenvironment in tumors. As ion channels play key roles in the physiology/pathophysiology of immune cells, we aimed at studying the ion channel repertoire in tumor-derived polymorphonuclear (PMN-MDSC) and monocytic (Mo-MDSC) MDSCs. Subcutaneous tumors in mice were induced by the Lewis lung carcinoma cell line (LLC). The presence of PMN-MDSC (CD11b^+^/Ly6G^+^) and Mo-MDSCs (CD11b^+^/Ly6C^+^) in the tumor tissue was confirmed using immunofluorescence microscopy and cells were identified as CD11b^+^/Ly6G^+^ PMN-MDSCs and CD11b^+^/Ly6C^+^/F4/80^−^/MHCII^−^ Mo-MDSCs using flow cytometry and sorting. The majority of the myeloid cells infiltrating the LLC tumors were PMN-MDSC (~60%) as compared to ~10% being Mo-MDSCs. We showed that PMN- and Mo-MDSCs express the Hv1 H^+^ channel both at the mRNA and at the protein level and that the biophysical and pharmacological properties of the whole-cell currents recapitulate the hallmarks of Hv1 currents: ~40 mV shift in the activation threshold of the current per unit change in the extracellular pH, high H^+^ selectivity, and sensitivity to the Hv1 inhibitor ClGBI. As MDSCs exert immunosuppression mainly by producing reactive oxygen species which is coupled to Hv1-mediated H^+^ currents, Hv1 might be an attractive target for inhibition of MDSCs in tumors.

## 1. Introduction

The tumor microenvironment (TME) is composed of a complex mixture of tumor-associated fibroblasts, infiltrating immune cells, endothelial cells, extracellular matrix proteins (ECM), proteases and signaling molecules, as well as cytokines [1]. Interactions between these cellular and acellular constituents of the microenvironment play critical roles in cancer development and the response to therapeutics [1]. The immune system plays a crucial role in the regulation of progression of multiple tumor types [2]. Tumors develop strategies to escape the immune attack in order to survive. At early stages of tumor development, the immune cells efficiently remove the transformed cells; however, this function becomes ineffective as cancer progresses [2].

Myeloid-derived suppressor cells (MDSCs) represent heterogeneous, phenotypically immature myeloid cells that play a tumor-promoting role by maintaining a state of immunologic anergy and tolerance [3]. Activated MDSCs secrete chemokines, cytokines, and enzymes, which suppress local T-cell activation and viability [3]. In addition, MDSCs can suppress the anti-cancer effect of T cells through deprivation of nutrients, such as l-arginine and l-cysteine in the TME. A key characteristic of MDSCs is their generation of reactive oxygen (ROS) [4] and reactive nitrogen species (RNS) in the TME [5]. Consequently, the T cell receptor becomes oxidized and loses its ability to recognize foreign antigens. Moreover, they influence the chemotactic activity of T cells and this impairs the recruitment of cytotoxic CD8^+^ T cells to the TME [3,6]. MDSCs boost a group of T cells, the regulatory T cells (T regs), that are important for immune tolerance [7,8]. Two distinct subsets of MDSCs, polymorphonuclear (PMN-MDSCs) and monocytic (Mo-MDSCs), can be identified, each promoting tumor progression by different mechanisms and to a different extent [9]. Selective elimination of PMN-MDSCs is sufficient to induce the activation and proliferation of systemic and intra-tumor CD8^+^ T cells [10]. Owing to their versatile immunosuppressive effects, MDSCs represent an attractive but somewhat elusive potential therapeutic target.

LLC, a murine tumor model for non-small cell lung carcinoma [11], is known to host a very low number of CD4^+^ and CD8^+^ T lymphocytes and a large number of myeloid cells including macrophages, neutrophils, and dendritic cells. Because of this, it is considered an immunologically “cold” tumor [12]. It has been observed that MDSCs accumulate in large numbers both in the spleen [13,14] and in the blood [15] of LLC tumor-bearing mice. Compared to other murine cell lines of lung carcinoma, LLC tumors are characterized by a significantly lower infiltration of CD3^+^ T lymphocytes, in particular, the anti-tumoral cytotoxic CD8^+^ subgroup. This “cold” environment not only drives MDSCs into the tumor but renders immune checkpoint inhibitors like anti-PD1 and anti-PD-L1 to be almost completely ineffective in slowing cancer progression [16]. Therapies inhibiting MDSCs enhance the therapeutic effects of anti-PD1 antibodies [17].

Ion channels and transporters are well-known regulators of effector functions of T and B lymphocyte such as cytokine production, proliferation, differentiation, and cytotoxicity [18]. However, little is known about the expression of ion channels and their activity in MDSCs. Until recently, two types of ion channels have been described in MDSCs: P2X7R and TRPV1. In a murine neuroblastoma model, P2X7R is expressed in both spleen Mo- and PMN-MDSCs, but with different localization and function. Mo-MDSCs produce more arginase-1, TGF-β1, ROS, and upon activation with ATP, they secrete more CCL2, a tumor-promoting cytokine, than their neutrophil-like counterparts [19]. TRPV1 activation with cannabidiol stimulates the recruitment and activation of MDSCs and this exerts a protective function in a murine hepatitis model [20]. Very recently, the functional expression of the Hv1 proton channel was described in in vitro differentiated murine MDSCs [21].

The Hv1 proton channel is built up by a voltage sensor domain, an intracellular C- and N-terminal domain and the proton transport occurs through the voltage sensor domain since it lacks a classical pore [22]. Hv1 is activated, among others, by membrane depolarization, the pH gradient across the plasma membrane and temperature [23]. The Hv1 proton channel has been described in inflammatory cells such as granulocytes [24,25] macrophages [26], eosinophils [27], B cells [28], plasmacytoid DCs [29] and to a less extent, in T cells [30,31]. The Hv1 proton channel has been shown to regulate intracellular pH of tumor cells by mediating outward H^+^ fluxes, thereby contributing to the acidification of the TME and the enhanced survival and mobility of tumor cells [32,33]. Pharmacological blocking of Hv1 induces intracellular acidification, which leads to apoptosis [34].

While acidic TME promotes survival and proliferation of cancer cells, it impairs the function of effector T cells [35]. Hv1 proton channel has been shown to support ROS production through NADPH oxidase activity in immune cells of innate and adaptive immunity [36]. However, so far, there are no data about the Hv1 proton channel in tumor-associated inflammatory cells, including MDSCs. In the present study, we aimed to investigate the Hv1 proton channel in MDSCs associated with Lewis lung carcinoma. Our results show that both Mo-MDSCs and PMN-MDSCs isolated from the tumor express Hv1 proton channel at RNA and protein level. Using whole-cell patch clamping, we demonstrated proton currents in both tumor-derived MDSCs, which could be blocked by the proton channel inhibitor 5-chloro-2-guanidinobenzimidazole (ClGBI).

## 2. Results

### 2.1. Identification, Isolation, and Functional Characterization of Tumor-Associated Mo- and PMN-MDCs

An increased number of MDSCs is correlated with bad/worse prognosis in multiple tumor types [3]. First, we aimed to visualize the presence and localization of MDSC subsets within the TME of the LLC lung tumors induced in mice. MDSCs were identified by the co-expression of CD11b (myeloid cell-specific marker) and either Ly6G (granulocytic marker, PMN-MDSCs) or Ly6C (monocytic marker, Mo-MDSCs) using immunofluorescence (Figure 1). PMN-MDSC-like cells represented an abundant group of cells and formed extensive aggregates in the tumor tissue (Figure 1A). The phenotypically Mo-MDSCs were scattered throughout the tumor sections and, however, were less abundant than PMN-MDSCs (Figure 1B). Additionally, we observed blood vessel staining with the Ly6C antibody, similar to recently published data [37].

Next, MDSCs were isolated from the TMEs of LLC-tumor-bearing mice according to Materials and Methods and MDSC subfractions were sorted using flow cytometry (Figure 2A). First, we gated for all viable myeloid cells using a combination of morphology, singlet, and viability gates in combination with the myeloid marker CD11b^+^ (Figure 2(A1–A4)). Within this population, we identified PMN-MDSCs as Ly6G^+^ cells (Figure 2(A5)), and Mo-MDSCs were identified as CD11b^+^/Ly6G^−^/F4/80^−^/MHCII^−^/Ly6C^+^ cells (Figure 2(A6,A7)). We determined the relative proportion of these two populations within the CD11b^+^ cells in the tumor and detected a predominance of PMN-MDSCs (60%) compared to Mo-MDSCs (around 10%) (Figure 2B). The remaining 30% of myeloid cells include tumor-associated dendritic cells (TADC), macrophages (TAM), and MDSCs differentiating into TAMs. However, even these MDSC cell surface markers have overlapping expression patterns with other cell types such as monocytes and granulocytes and to date, no specific marker combinations have been described that unequivocally identify MDSC sub-populations [38,39].

The main characteristic of MDSCs Is their potent immune suppressive nature. To determine if purified MDSCs display immunosuppressive potency, we set up a polyclonal T cell proliferation suppression assay. Under these experimental conditions, Mo-MDSCs had suppressive capacity when they were co-cultured with murine splenocytes in 1:1 and 1:2 ratios, whereas PMN-MDCSs failed to demonstrate immunosuppressive properties in accordance with other studies [9]. Mo-MDSCs were the dominant immunosuppressive population of these MDSCs that suppressed both CD8^+^ and CD4^+^ T cell proliferation (Figure 3 and Figure 4).

### 2.2. Detection of Hv1 Proton Channel in Tumor Derived MDSCs

A recent study reported the expression of the voltage-gated proton channel Hv1 in in vitro differentiated MDSCs [21]. Motivated by this, we tested the expression of the Hv1 transcript in PMN- and Mo-MDSCs isolated from the LLC tumor using qPCR (Figure 5). Albeit both MDSCs subpopulations expressed the Hv1 gene, we found that the expression level in PMN-MDSCs was higher than in Mo-MDSCs. Western blot analysis using an antibody specific for Hv1 showed the same pattern, a higher Hv1 protein level in PMN-MDSCs compared to Mo-MDSCs.

### 2.3. Expression of Hv1 in MDSCs Infiltrating LLC Tumors

To gain more insight into the in situ expression of Hv1 in MDSCs infiltrating the LLC tumors, we performed immunofluorescence staining of LLC tumor sections using the Hv1-specific antibody validated in WB analysis. Figure 6A shows strong Hv1 immunofluorescence that overlaps with the expression of the myeloid marker CD11b. As expected, based on the WB data, the Hv1 signal was much stronger in PMN-MDSCs (Figure 6B) compared to Mo-MDSCs (Figure 6C). PMN-MDSCs in the tumor showed focal distribution whereas the Mo-DSCSs are more sparsely distributed, similar to the images in Figure 1.

### 2.4. Ion Currents in MDSCs

MDSCs obtained by cell sorting of the tumoral mass were analyzed using single-cell electrophysiology (patch-clamp) for the expression of whole-cell ion currents using various intra- and extracellular solution combinations. Initially, we used K^+^-based intracellular and Na^+^-based extracellular solutions, which allow the recording of voltage-gated K^+^ and Na^+^ currents (see Materials and Methods). When either PMN- or Mo-MDSCs were subjected to 15 ms-long voltage steps from −100 mV holding potential to +50 mV test potential, we could not detect any classical voltage-gated ion currents in the outward direction (Appendix A). Ion currents over a wider range of membrane potentials and depolarization durations were studied using voltage ramps. In these experiments (Appendix A), we did not see inward currents characteristic of the presence of voltage-gated Na^+^ or Ca^2+^ channels. We did not optimize further the ion concentrations and voltage protocols for recording Na^+^ and Ca^2+^ currents, so we cannot exclude the possibility that a more detailed analysis would report some currents that were not readily seen during the initial characterization of the MDSC currents. On the other hand, we detected a voltage-gated outward current that activated at depolarized membrane potentials and was sensitive to the extracellular pH (Appendix A). The presence of the proton current was observed more clearly when the recording solutions lacked conventional permeating cations and contained reduced Cl^−^ concentration to eliminate outward currents other than the proton current (using NMDG-based solutions) and also rich in non-volatile buffers in order to keep both pH_i_ and pH_e_ stable [40].

Figure 7 shows the instantaneous I-V curves obtained using a voltage ramp protocol (from −60 mV to +150 mV), while keeping the intracellular pH constant at 6.2 and changing the extracellular pH from 5.7 to 7.4 in 3 steps. The Hv1 channel is characterized by a phenomenon called ΔpH-dependent gating: a change in the intracellular pH or in the extracellular pH strongly modulates the voltage at which the channel opens (threshold voltage, V_thr_) [41]. Figure 7A,B show several features characteristic of Hv1. First, the smaller the pH gradient across the membrane [ΔpH_e–i_ = (pH_e_ − pH_i_)], the more depolarized the V_thr_. Second: the larger the pH gradient, the larger the currents are at identical membrane potentials. The qualitative and quantitative analysis of the latter phenomenon is in Figure 7C–F. Plotting the current of each sweep at +145 mV as a function of the sequentially numbered sweep numbers (the time interval between the sweeps is 15 s) shows that changing the extracellular pH induces rapid and reversible effects on the current amplitude with currents being gradually smaller as the extracellular pH becomes more acidic, for both PMN-MDSCs (Figure 7C) and Mo-MDSCs (Figure 7D). Even if the capacitance measurements suggest that Mo-MDSCs are bigger than PMN-MDSCs (1.83 ± 0.14 (*n* = 40) vs. 3.13 ± 0.14 pF (*n* = 39), mean ± SEM, *p* < 0.0001) (Appendix A), the H^+^ current density in PMN-MDSCs (Figure 7E) was ~3 times bigger compared to Mo-MDSCs at pH_e_ = 7.4 (Figure 7F). At every pH_e_ value, except 5.7, the current density on PMN-MDSCs was significantly larger than on Mo-MDSCs.

The V_thr_ shift can be quantitatively inferred from the current–voltage relationship shown in Figure 8. The families of whole-cell currents in Figure 8A–D were obtained in a single PMN-MDSC upon applying 2000 ms-long step voltage depolarizations in 10 mV increments. The intracellular pH was maintained at pH_i_ = 6.2 and the extracellular pH ranged from pH_e_ = 7.4 (Figure 8A) to pH_e_ = 5.7 (Figure 8D). Comparison of the topmost traces in Figure 8 panels A–D, obtained at +100 mV, indicates that the larger the pH gradient, the larger the current and the quicker its activation kinetics. Although larger depolarizations caused currents with faster activation kinetics, 2 s long pulses were not long enough to obtain saturation of the current. This can be observed both in PMN-MDSCs (Figure 8A–D) as well as in Mo-MDSCs (Appendix A) and it is a common feature of Hv1 currents [40,42]). Figure 8E,F show the normalized peak currents as function of the membrane potential. The V_thr_ values were determined using the statistical criteria explained in detail in the Materials and Methods and in [42] and indicated by arrows in Figure 8E for PMN-MDSCs and in Figure 8F for Mo-MDSCs. The V_thr_ values are shifted to more depolarized membrane potentials as pH_e_ became more acidic while keeping the intracellular pH at pH_i_ = 6.2. For statistical analysis, the V_thr_ values were individually determined on a cell-by-cell basis and plotted as a function of the extracellular pH in Figure 9A, and as a function of ΔpH in Figure 9B. The numerical values of V_thr_ for PMN-MDSCs were −16.7 ± 3.1 mV at ΔpH_e–i_ = 1.2; +20 ± 5 mV at ΔpH_e–i_ = 0.2; +15.0 ± 6.5 mV at ΔpH_e–i_ = 0; and +63.3 ± 3.3 mV at ΔpH_e–i_ = −0.5. For Mo-MDSCs, the V_thr_ was −14.4 ± 5.5 mV for ΔpH_e–i_ = 1.2; +40.0 ± 10.8 mV at ΔpH_e–i_ = 0.2, +38.0 ± 9.2 mV for ΔpH_e–i_ = 0; and +57.5 ± 7.5 mV at ΔpH_e–i_ = −0.5 (Figure 9A). These values were used to indicate the V_thr_ in the current–voltage relationships (Figure 8E,F) as colored arrows. The linear regression analysis of the V_thr_-ΔpH_e–i_ relationship did not deviate from the “rule of forty”, i.e., (~40 mV shift per one unit ΔpH change) for either PMN- or Mo-MDSCs [41] (Figure 9B); however, the V_thr_-ΔpH_e–i_ is shifted to depolarized potentials for the currents recorded in Mo-MDSCs (Figure 9A). This is indicated by a slight upward shift in the V_thr_-ΔpH_e–i_ relationship obtained for Mo-MDSCs (dashed line in Figure 9B) versus that for PMN-MDSCs (solid line in Figure 9B), i.e., at identical pH gradients the thresholds are more positive for Mo-MDSCs as compared to PMN-MDSCs.

High H^+^ selectivity is a prominent property of the Hv1 channels [43]. The H^+^ selectivity of the currents in PMN-MDSCs was estimated by determining the reversal potential (E_rev_) from the analysis of the whole-cell tail currents. The cells were depolarized to +100 mV for 500 ms to activate the current, followed by repolarizations to various membrane potentials (from −60 to +100 mV) to obtain the tail currents. Figure 10A–D shows a representative set of the tail current experiments at pH_i_ 6.2 and at the indicated pH_e_ values ranging from 5.7 to 7.4. The membrane potential at which the tail current reversed its polarity was considered as the reversal potential. Figure 10E shows selected traces at higher time and amplitude resolutions to illustrate the determination of E_rev_. As expected for a H^+^ current, the E_rev_ approached ~0 mV at ΔpH_e–i_ = 0 (−1.3 ± 4.3 mV, mean ± SEM, *n* = 11). The E_rev_ values were plotted against ΔpH_e–i_ in Figure 10F and a straight line was fit to the data points. The slope of the best linear regression line indicates that E_rev_ shifts −42 mV for every one-unit shift in the extracellular pH (Figure 10F). However, the slope is different from a perfectly selective H^+^ conductance (−59.16 mV/ΔpH) as predicted by the Nernst equation. Mo-MDSCs’ currents were too low for a reliable tail current analysis.

ClGBI is an apolar small molecule, which can cross the plasma membrane and block the Hv1 channel from the cytosolic side [44]. The sensitivity to guanidine derivatives, and particularly to ClGBI, is often used in the literature as a pharmacological argument for the identification of Hv1 currents [21,34,42,45]. Accordingly, we tested the sensitivity of the whole-cell currents to ClGBI at 200 µM concentration, which was reported to block ~80% of the Hv1 current in a reversible manner [44]. Figure 11A shows that 200 µM ClGBI blocked almost completely the whole cell current in a PMN-MDSC and that the block was reversible, the current returned to the control upon washing the recording chamber with the ClGBI-free extracellular solution. The development of the block was very fast whereas >30 episodes in a ClGBI-free solution were needed to wash-out the effect (Figure 11B). While PMN-MDSCs were robust enough to withstand repeated 2000 ms-long depolarizations from −80 mV to +100 mV (Figure 11A), we were able to do pharmacological experiments in Mo-MDSCs using repeated application of a voltage-ramp protocol where long exposure to depolarized test potentials can be avoided (Figure 11C). Regardless of the voltage protocol used (i.e., step depolarization vs. voltage ramp), the application of 200 µM ClGBI reduced the magnitude of the current significantly, ~80% reduction in PMN-MDSCs and 75% in Mo-MDSCs (Figure 11D).

## 3. Discussion

Our paper demonstrates for the first time that murine MDSCs obtained directly from tumor tissue express the Hv1 H^+^ channel both at the mRNA and at the protein level and that the properties of the whole-cell current in tumor-derived MDSCs recapitulate the hallmarks of Hv1 currents recorded in various cells and in cells expressing Hv1 heterologously. These hallmarks are the voltage-dependent activation, ~40 mV shift in the activation threshold of the current per unit change in the extracellular pH and high H^+^ selectivity as reviewed extensively by [43]; and the sensitivity to the guanidine derivative ClGBI [44].

A key novelty of our study is that the expression of Hv1 was shown in MDSCs obtained from tumors. As described in the introduction, the LLC tumor of mice was a good candidate for the isolation of MDSCs. LLC is considered an immunologically “cold” tumor [12] which is characterized by high MDSC infiltration and an immunosuppressive microenvironment [46]. Although MDSCs are in the focus of tumor immunology and are intensively investigated there is no consensus regarding the phenotypic definition of these cells. Usually, in mice, Mo-MDSCs are defined as CD11b^+^Ly6C^+^ and PMN-MDSCs as CD11b^+^Ly6G^+^Ly6C^low^, but these markers are commonly defining other subsets of myeloid cells as well [47]. Taking into consideration the limitation of the identification of these cells by cell surface markers, we demonstrated the presence of both PMN-MDCSs and Mo-MDSCs in LLC tumors induced in mice using immunofluorescence (Figure 1). Moreover, Mo-MDSCs can be distinguished from tumor-associated macrophages (TAMs) because of their lower expression of F4/80 [48]. This was utilized in flow cytometric separation and specific enrichment MDSCs for electrophysiological investigations. We found tumor-derived PMN-MDSCs to be the most abundant subset in LLC, being detected by flow cytometry ~6 times more than Mo-MDSCs (Figure 2B). This is common for this kind of tumors [49] and similar proportions have been reported in pancreatic ductal adenocarcinoma (PDAC) [50] and autoimmune diseases like autoimmune arthritis as well [51].

Since the definition of MDSCs via membrane markers is not straightforward, it is common practice to verify the identity of the cells with a functional study. This is usually achieved by demonstrating the suppression of T cell proliferation to avoid confusion with phenotypically similar monocytes and neutrophils [48]. In our hands, tumor-derived Mo-MDSCs were able to suppress T-cell proliferation, while PMN-MDSCs, although more abundant, did not suppress T-cells, at least at the MDSC/splenocyte ratios we used. This insufficient anti-proliferative phenotype of PMN-MDSCs agrees with other studies performed with LLC tumors in mice [9] and it has been observed in PDAC [52], autoimmune arthritis [51], and MDSCs accumulating in transplanted organs in humans [53]. However, PMN-MDSCs may promote tumor growth independent of the inhibition of T cell proliferation by directly inhibiting cytolytic T cell activation and indirectly influencing other myeloid cells and NK cells [54,55]. PMN-MDSCs are the major source of immunosuppressive mediators like ROS and RNS, which suppress TCR signaling and modulate cytokine secretion [56]. Additionally, PMN-MDSCs impair recruitment of cytolytic T cells [57] and contribute the tumor progression by secreting MMPs and factors that promote tumor angiogenesis [58,59,60]. Moreover, a recent electrophysiological study strongly supports that the cells we classified as PMN-MDSCs are different from neutrophils. Using electrophysiological assays, Immler and co-workers have shown that neutrophil polymorphonuclear leukocytes (PMN) functionally express voltage-gated Kv1.3 K^+^ channels [61], in clear contrast to our whole-cell records, where Kv1.3 or any other voltage-gated K^+^ current was missing.

Several lines of evidence support that whole-cell Hv1 currents were recorded in tumor-derived Mo- and PMN-MDSCs in our study. First, the currents were slowly activating, rapidly deactivating, and with no sign of inactivation, which is characteristic of Hv1. Moreover, the currents were recorded using intra- and extracellular solutions that lacked (K^+^, Na^+^) or contained negligible concentration (Cl^−^) of conventional permeating ions; thus, the contribution of other conductances to the whole-cell current, that could mimic the behavior of Hv1, are minimized.

Second, the whole-cell currents in both MDSC types were sensitive to the pH gradient across the membrane and the membrane potential. The threshold voltage for the activation of the currents shifts along the voltage axis when changing ΔpH, ~40 mV per unit change in the extracellular pH, closely mirroring what has already been described for proton currents in various cells [43,62] including bone-marrow-derived MDSCs [21]. On the other hand, V_thr_ of the current varies among different cell types [42,63]. For example, at identical pH gradients (ΔpH = 1.2) and recording solutions, the Hv1 current in human chorion-derived mesenchymal cells activate at ~10 mV more positive membrane potential than hHv1 expressed in HEK-293 cells [42], whereas the V_thr_ values in both types of MDSCs in this study are ~ 10 mV more negative than that of hHv1 in HEK-293. This more negative threshold potential may facilitate the opening of Hv1 at membrane potentials in the physiological range typically observed for non-excitable cells. Nevertheless, the V_thr_ values determined in our study are more positive than the reversal potentials of the H^+^ currents obtained at a wide range of ΔpH values (compare Figure 9 and Figure 10), thus, allowing the Hv1 channel to conduct protons solely in the outward direction, similar to other cells [43]. We also found that the V_thr_ in PMN-MDSCs is depolarized as compared to Mo-MDSCs under symmetrical pH conditions (pH_e_ ~ pH_i_), even if this did not affect the overall V_thr–_ΔpH relationship of –40 mV/ΔpH [64]. The V_thr_ of ~+40 mV in symmetric solutions in PMN-MDSCs is qualitatively similar to what has been determined for the Hv1 current in murine neutrophil granulocytes (~+50 mV, [65]), that are closely related to PMN-MDSCs. Moreover, specific mutations generated in the hHv1 channel drastically modify V_thr_, without influencing the V_thr_-ΔpH relationship [64]. We do not know whether our observations originate from technical errors mainly due to the extremely low ion currents in Mo-MDSCs or from a translational or post-translational difference between Hv1 in PMN- and Mo-MDSCs.

Third, the Hv1 current in PMN-MDSCs is fairly H^+^-selective, since the E_rev_-ΔpH relationship resembles the theoretical relationship obtained for H^+^ from the Nernst equation. The slope of the E_rev_-ΔpH relationship in rat alveolar epithelial cells [62] and canine myocytes [66] is similar to the theoretical slope calculated from the Nernst equation for H^+^ (~−59 mV/ΔpH); however, slopes in Jurkat (−47 mV/ΔpH, [30]) and in MDSCs in our study (−42 mV/ΔpH) are shallower. A similar discrepancy has been observed in murine microglia as well, where proton depletion, as a consequence of the proton current passing through Hv1 channels, was suggested to account for the shallower slope [67]. In addition, the small currents in MDSCs can be easily contaminated by non-specific leak even if leak corrections are applied: any contribution of leak to the whole cell current shifts the reversal potentials to depolarized potentials. The complications originating from incomplete leak subtraction ruled out the reliable determination of the reversal potential in Mo-MDSCS where currents are very small, in many cases less than 100 pA even under optimal ΔpH and membrane potential combinations.

Fourth, the currents in both PMN-MDSCs and Mo-MDSCs were sensitive to ClGBI, the guanidine derivative small molecule inhibitor Hv1 [44]. Even if the selectivity of ClGBI towards Hv1 has not been assessed yet, it is widely used as an indicator of the presence of the Hv1 current in various cell types [42,66,68]. We showed that ClGBI at 200 μM concentration reversibly blocks ~80% of the whole cell currents in both PMN- and Mo-MDSCS. Based on the block percentage and assuming a sigmoidal dose-response function with a Hill coefficient of 1, the single-point estimate of the IC_50_ is ~50 μM, which is consistent with the reported potency of ClGBI in inhibiting Hv1 [44,66]. This pharmacological clue strengthens our conclusion that these currents correspond to proton currents mediated by Hv1 in MDSCs.

Fifth, the electrophysiological data are strongly supported by molecular biology where the mRNA transcript of Hv1 was identified in MDSCs using RT-qPCR along with the Hv1 protein itself in Western blots. The human Hv1 proton channel protein has two isoforms, a long, full-length isoform and a short one, which lacks an N-terminal region due to alternative splicing [69]. The antibody used in the present study recognizes both isoforms, which is confirmed in the CH12 mouse B cell lymphoma cell line used in our study as a positive control, and similar to human B cell lymphomas described previously [69]. However, we detected only the long form both in PMN-MDCS and Mo-MDSCs isolated from the LLC tumor. The short form might not be expressed, or it is in a negligible amount, under the detection limit, suggesting that the long isoform of Hv1 may contribute to the function of tumor-derived MDSCs.

Although the Hv1 proton channel has been well characterized in several immune cell types, to our knowledge, there is no information about Hv1 expression in tumor-associated inflammatory cells. Our study, for the first time, detected within a tumor a high number of Hv1^+^ myeloid cells that are consistent with the cell surface maker phenotype of PMN- and Mo-MDSCs (see above). Our electrophysiology results are consistent with the data described for in vitro produced MDSCs where Hv1 H^+^ currents of similar magnitude (between 200 pA and 1 nA at +130 mV) to our study were reported using patch-clamp in a mixed MDSC population [21]. The MDSCs used by Alvear-Arias et al. were obtained by induction of the differentiation of bone marrow-derived myeloid precursors by GM-CSF [21] whereas in our study, MDSCs were isolated directly from LLC tumors induced in mice. Our study suggests that in vitro differentiated MDSCs may serve as useful tools to understand Hv1-dependent regulation of T cell function in cancer as the channel phenotype of these cells is similar to the tumor-derived MDSCs. Moreover, we also showed that both PMN- and Mo-MDSCs display Hv1-mediated H^+^ currents, albeit to a different extent, so the ion channel phenotype of the two MDSCs subtypes is similar, at least in mice.

What can be the functional consequence of the Hv1-mediated H^+^ currents in MDSCs? Neutrophils, which are closely related to MDSCs, express a functional Hv1 and the Hv1 H^+^ currents contribute to the counterbalancing positive charge efflux required for the maintenance of ROS production [36,70]. As ROS production is also a hallmark of MDSCs’ immunosuppression [4], the functional expression of Hv1 in MDSCs and its sensitivity to Hv1 inhibitors seems logical. Consistent with this, Hv1-mediated H^+^ currents were shown in MDSCs using electrophysiology ([21] and this study), and blocking Hv1 using ClGBI and Zn^2+^ inhibited ROS production and alleviated the inhibition of T cell proliferation by MDSCs [21]. Longer than 2 h exposure of MDSCs to Hv1 inhibitors induced significant cell death which raised some ambiguity regarding the specificity of the effect of ClGBI application. This, and the application of a reversible blocker (ClGBI) in a short-term pre-incubation to MDSCs followed by wash-out, makes the interpretation of the data on T cell/MDSCs co-cultures complex and difficult. In our hands, ClGBI applied alone, in the absence of MDSCs, inhibited T cell proliferation, which ruled out T-cell/MDSC co-culture experiments in the presence of ClGBI. The potential side effects of currently available Hv1 inhibitors argue for the development of more specific and higher affinity Hv1 inhibitors.

The proton efflux through the Hv1 proton channel may also contribute to the acidic milieu in the TME, which is well tolerated by tumor cells but impairs the tumor suppressive ability of T cells, NK cells [35]. Thus, modulating the acidic tumor microenvironment by Hv1 inhibition may facilitate the tumor-suppressive effect of immune cells in cancer therapy. However, recently it has also been shown that the increase of intracellular acidity in activated T cells due to the lack of Hv1 proton channel reduces the effector function of T cells [31], which must also be considered to determine the overall outcome of Hv1-targeted cancer therapy.

## 4. Materials and Methods

### 4.1. Mice

C57BL/6J male mice were purchased from The Jackson Laboratory. Animals were housed under specific pathogen-free conditions and the experiments were carried out under Committee of Animal Research of the University of Debrecen institutional ethical guidelines and licenses (license number: 8/2014/ DEMÁB).

### 4.2. Cells

CH12 B cell lymphoma cells were grown in RPMI 1640 medium supplemented with 10% FBS, 1% GlutaMAX, 1% penicillin-streptomycin. The Lewis lung carcinoma cell line (LLC) was a kind gifts from László Nagy (Department of Biochemistry and Molecular Biology, Faculty of Medicine, University of Debrecen, Hungary). Cells were grown in RPMI 1640 medium supplemented with 10% FBS, 1% GlutaMAX, and 1% penicillin-streptomycin.

### 4.3. Tumor Model and Cell Sorting

LLC cells (3 × 10^6^/0.1 mL PBS) were injected subcutaneously on the right flank of syngeneic 8- to 12-week-old male C57BL/10 wild-type mice. Tumors were excised and cut into small pieces in isolation buffer (RPMI 1640 medium) followed by protease digestion using enzyme mixture (collagenase I (CLSS-1, Worthington, Columbus, OH, USA, 10 U/mL), collagenase IV (CLSS-4, Worthington, 400 U/mL), and DNase I (DCLS, Worthington, 30 U/mL)) for 30 min at 37 °C. Cells were filtered through a 70 μm strainer and the red blood cells were lysed in ACK (Ammonium Chloride Potassium) lysing buffer. After washing and spinning, cells were resuspended in Hanks’ Balanced Salt Solution (HBSS) buffer and stained with live/dead stain (Fixable Viability Dye eFluor 506, eBiocience, San Diego, CA, USA, 1 μg/mL) in HBSS for 30 min at 4 °C to exclude the dead cells. After washing, cells were preincubated in MACS buffer (phosphate buffered saline (PBS) pH7.2, supplemented with 0.5% BSA and 2 mM EDTA) with rat anti-mouse CD16/CD32 FcγR blocking antibody (Table 1, #1) for 5 min at 4 °C and stained for 30 min at 4 °C with antibodies #2,5,6,8,9, listed in Table 1 (FACS application), in order to sort the PMN-MDSCs and Mo-MDSCs, using BD FACSAria III sorter (BD Biosciences, San Hose, CA, USA).

### 4.4. Suppression of Polyclonal T Cell Proliferation

Mice were sacrificed by cervical dislocation and the spleen was excised and placed in RPMI 1640 medium. Splenocytes were obtained by pressing gently the spleen tissue through a 70 μm strainer using a 1 mL syringe plunger. The red blood cells were lysed in ACK buffer. Splenocytes were labeled using carboxyfluorescein succinimidyl ester (CFSE 1 µM; Sigma Aldrich/Merck KGaA, Darmstadt, Germany) for 30 min at 37 °C, then cultured with or without sorted PMN-MDSCs and Mo-MDSCs, respectively, in 1:1, 1:2, 1:4 (MDSC:splenocytes) ratio. Co-cultures were carried out in ME medium (RPMI 1480, 10% FBS, 1% GlutaMAX, 1% penicillin-streptomycin, 1 mM sodium pyruvate, 1× non-essential amino acids, and 48 nM 2-betamercaptoethanol) supplemented with IL-2 (Peprotech, 20 ng/mL) and anti-mouse CD3 (as a kind gift from Jo A. Van Ginderachter, Myeloid Cell Immunology Lab, VIB, Belgium, 1 µg/mL, #10 in Table 1) in 48-well plates for 2 days at 37 °C. Cells were collected, blocked using rat anti-mouse CD16/CD32 FcγR blocking antibody (#1 in Table 1), and stained with CD4 (#11), CD8 (#12), and CD11b-specific antibodies (#4) to mark CD4, CD8 T cells, and myeloid cells, respectively (see Table 1). T cell proliferation was measured by the extent of CFSE dilution using ACEA NovoCyte 2000R cytometer (Agilent, Santa Clara, CA, USA) as described previously [71].

### 4.5. Real Time Quantitative PCR (RT-qPCR)

Total RNA was isolated from the sorted cells using Trizol reagent (Life Technologies/Thermo Fisher Scientific, Waltham, MA, USA), treated with RNAse-free DNase 1 (Thermofisher, AM2222) for 30 min at 37 °C, followed by inactivation of the DNase at 75 °C for 10 min. The DNase-treated RNAs were submitted to cDNA synthesis. cDNAs were synthesized using a High-Capacity cDNA Reverse Transcription Kit (ThermoFischer Scientific, Waltham, MA, USA, cat #4368814) following the manufacturer’s instruction.

Quantitative PCR was performed by Applied Biosystems Step One Plus platform, 95 °C for 10 min, 40 cycles of 95 °C 15 sec and 60 °C 45 sec, using Light Cycler 480 SYBR Green I. Master mix (Roche, Basel, Switzerland). Gene expression was quantified by the comparative threshold cycle method and normalized to mouse 18S expression as a housekeeping gene. All PCR reactions were performed in duplicate. Values are expressed as means ± SD. The following primers were used: mouse *Hv1* (forward: TCGTGCTTGCTGAACTCCTCCT and reverse: GGCAAAGCTCATGTAGTGGAACG); *mouse 18S RNA*: (forward: GGGAGCCTGAGAAACGGC and reverse: GGGTCGGGAGTGGGTAATTTG).

### 4.6. Western Blot

The sorted cells were lysed in 2xLaemmli buffer supplemented with proteinase (Sigma/Merck KGaA, Darmstadt, Germany, cat #P8340) and phosphatase inhibitor (Sigma, cat #P5726) cocktail followed by a denaturation step at 98 °C for 5 min. Fifteen µg protein per sample was run on 7.5% polyacrylamide gel and transferred to a nitrocellulose membrane (Bio-Rad Laboratories, Hercules, CA, USA). The non-specific binding sites were blocked by incubating the nitrocellulose membrane in 5% *w*/*v* non-fat dry milk in Tris-Buffered Saline (TBS) supplemented with 0.2% *v*/*v* Tween (TBST) for 1 h. Next, the nitrocellulose membrane was incubated with primary antibodies (Hv1 (#13) and β-actin (#14), see Table 1) diluted in 2.5% *w*/*v* non-fat dry milk in TBST overnight at 4 °C. After washing for 3 × 7 min in TTBS, the nitrocellulose membrane was incubated with donkey anti-rabbit IgG-HRP linked (#17) secondary antibody for detection of Hv1 protein. Sheep anti-mouse IgG-HRP linked (#15) secondary antibody was used for the detection of actin. The chemiluminescence signal was detected using an Azure c300 Gel Imaging System (Azure Biosystems, Dublin, CA, USA). CH12 B cell lymphoma lysate was used as a positive control. The Western blot analysis was performed on sorted cells from 3 different experiments. The relative Hv1 protein level was calculated using Azure Spot Pro Analysis software and expressed as Hv1/actin ratio.

### 4.7. Immunofluorescence

Frozen sections were fixed in methanol at −20 °C for 15 min, washed, blocked in 3% bovine serum albumin in PBS salt solution, and incubated overnight at 4 °C with the antibodies diluted in blocking solution. The antibodies used for staining cryosections are listed in Table 1 (#3, 5, 7, 13).

After washing for 3 × 5 min in 1×PBS, sections stained for Hv1 were additionally incubated for 1 h at room temperature with an appropriate secondary antibody (#16). Sections were stained with DAPI solution (Invitrogen, 1 µg/mL) to visualize nuclei and mounted with Fluoromount G (eBioScience, San Diego, CA, USA) mounting media. The specificity of the secondary antibody was verified by omitting the primary antibody from the staining procedure. Sections were examined using a LSM800 microscope and Zen 2.3 SP1 software (Zeiss, Jena, Germany).

### 4.8. Electrophysiology and Pharmacology

Electrophysiology measurements were carried out using the patch-clamp technique in voltage-clamp mode. Whole-cell currents were recorded from murine PMN- and Mo-MDSCs using a Multiclamp 700B amplifier connected to a DigiData 1440A digitizer (Molecular Devices, Sunnyvale, CA, USA). Micropipettes were pulled from GC 150 F-15 borosilicate capillaries (Harvard Apparatus, Kent, UK) resulting in 3- to 5-MΩ resistance in the bath solution. The standard extracellular solution used to study Hv1 at pH_e_ = 7.4 contained 180 mM HEPES, 75 mM N-Methyl-D- Glucamine (NMDG), 15 mM glucose, and 3 mM MgCl_2_ (titrated with CsOH), whereas in the extracellular solutions at pH_e_ = 6.4/6.2/5.7, 180 mM HEPES buffer was substituted with 180 mM MES (2-(N-morpholino)ethanesulfonic acid, titrated with CsOH or HCl). The standard intracellular solution at pH_i_ = 6.2 contained 180 mM MES, 75 mM NMDG, 15 mM glucose, 3 mM MgCl_2,_ and 1 mM EGTA (ethylene glycol-bis(β-aminoethyl ether)-N,N,N’,N’-tetraacetic acid, titrated with CsOH). To explore the presence of other voltage-gated currents, we used a Na^+^-based extracellular solution at pH_e_ = 7.35 containing 145 mM NaCl, 5 mM KCl, 1 mM MgCl_2_, 2.5 mM CaCl_2_, 5.5 mM glucose, and 10 mM HEPES and a K^+^-based intracellular solution at pH_i_ = 7.22 containing 140 mM KF, 2 mM MgCl_2_, 1 mM CaCl_2_, 10 mM HEPES, and 11 mM EGTA. The pH of the solutions was checked before every experiment, and all salts and components of the solutions were purchased from Sigma-Aldrich Budapest, Hungary. A custom-built gravity-driven perfusion system was used to provide the necessary solution exchange around the cells.

The guanidine derivative Hv1 blocker 5-cholor-2-guaninidbenzimidazole (ClGBI, Sigma Aldrich Kft. Budapest, Hungary, S517038) was kept in DMSO at a stock concentration of 100 mM and suitably diluted in the standard extracellular solution when needed.

### 4.9. Electrophysiology Data Acquisition and Analysis

Voltage ramps (2000 ms long from −60 mV to +150 mV, every 15 s) were used to demonstrate qualitatively the dependence of the activation threshold of the Hv1 currents at various pH_e_ values. Traces were filtered (lowpass boxcar, 25 smoothing points), off-line leak-corrected manually point-by-point. Linear regression line was fit to the data points below the activation threshold of the H^+^ current (between 50 ms to 330 ms, corresponding to −60 mV and −30 mV) and the fitted parameters were used to subtract the non-specific leak [42]. The leak-corrected currents between +145 mV and +146 mV were extracted, averaged, and considered as the peak current. The average currents of two or three stable traces at a given pH_e_ condition defined one data point. Currents are either shown as absolute values or expressed as current density obtained by dividing the currents measured in pA with the cell capacitance in pF to yield pA/pF.

The current–voltage (I–V) relationships and the activation threshold voltage of the currents (V_thr_) were determined using 2 s long step depolarizations from a holding potential of −80 mV to +100 mV in +10 mV increments. The protocol was applied every 15 s; the sampling rate was 5 kHz. For the I-V curves, every trace was filtered (lowpass boxcar, 25 smoothing points) and leak-corrected manually. Peak currents were calculated as the average of the last 18 points (i.e., between 2051.1 and 2051.9 ms) at the end of the depolarizing pulses. For the V_thr_ determination, leak correction was performed using the first 5 (pH_e_ 7.4), 7 (pH_e_ 6.4 and 6.2), and 10 peak currents of the I-V relationship (i.e., between −80 and −50/−30/+10 mV) and the SD was calculated using the first 5 values in the I-V (i.e., between −80 and −50 mV). The V_thr_ was selected as the membrane potential at which the current was above 2 × SD [42].

For recording tail currents, the Hv1 current was fully activated using 500 ms long single-step depolarizations from a holding potential of −80 mV to +100 mV. The tail currents were recorded upon stepping back from this potential in 20 mV decrements to −60 mV, and the currents were recorded for 250 ms at the back-step potentials. The protocol was applied every 15 s with a sampling rate of 20 kHz. The traces were leak-corrected manually and filtered (lowpass boxcar, 25 smoothing points).

To test the presence of voltage-gated K^+^ currents, 15 ms long depolarization steps were applied to +50 mV from a holding potential of −100 mV every 15 s, with a sampling rate of 20 kHz. Voltage ramps, as specified above, were also used to study the presence of voltage-gated ion currents in MDSCs over an extended membrane potential range and depolarization duration in physiological salt solutions.

The pClamp 10.5, 10.7, and 11.2 software packages were used to acquire the data. The pClamp 10.7 and 11.1 software packages (Molecular Devices Inc., Sunnyvale, CA, USA) were used to analyze the data. Statistical analyses were performed with GraphPad Prism 8.4.3 (GraphPad Software, Inc., San Diego, CA, USA).

## Figures and Tables

**Figure 1 ijms-24-06216-f001:**
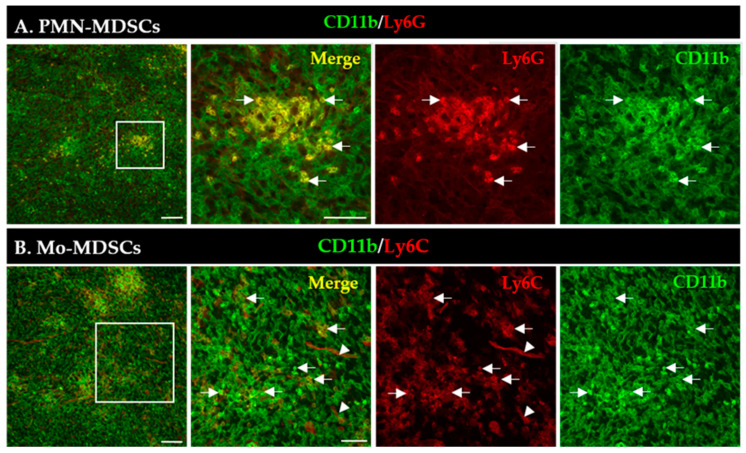
Detection of PMN-MDSCs (**A**) and Mo-MDSCs (**B**) cells in LLC cryosections by immunofluorescence. Staining with CD11b-specific antibody marks all myeloid cells (green), while co-staining with Ly6G (**A**) and Ly6C (**B**) antibodies (red) reveals PMN-MDSCs and Mo-MDSCs (arrows), respectively. Labels in the top right corner of the panels indicate the images obtained with filter settings specific for individual fluorophores or the merged image. Arrowheads mark blood vessels stained with the Ly6C antibody. Boxed areas in the left images are shown in higher magnification to the right panels. Scale bars are 100 µm for the low and 50 µm for higher magnification images.

**Figure 2 ijms-24-06216-f002:**
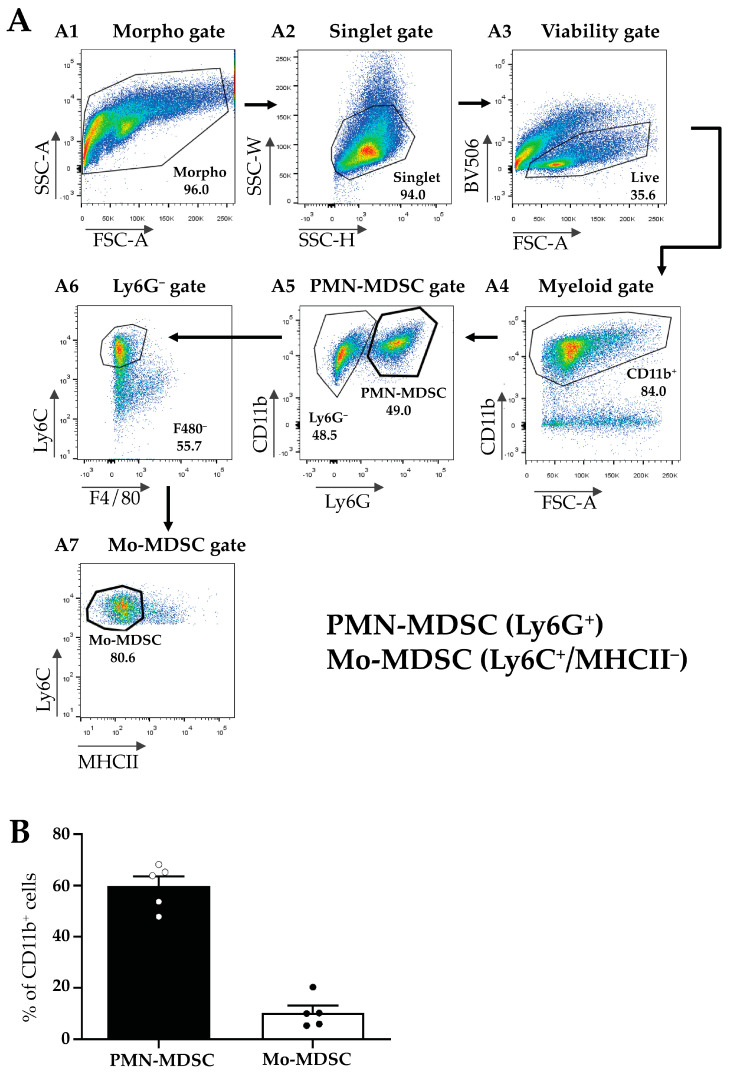
Flow cytometric characterization of the myeloid cellular populations in LLC tumors. (**A**) Gating strategy for the identification and sorting of myeloid subsets such as Mo-MDSCs and PMN-MDSCs from the TME. Panels show representative flow cytometric plots of the sorting of MDSC subsets. Following the application of the morphological gates ((**A1**) Morpho gate and (**A2**) Singlet gate) viable cells were sorted based on the Fixable Viability Dye eFluor 506 exclusion ((**A3**) Viability gate, BV 506). Myeloid cells were defined by their CD11b expression ((**A4**) Myeloid gate) and subsequently, distinguished into two subsets by Ly6G staining. Ly6G^+^ cells were sorted as PMN-MDSCs ((**A5**) PMN-MDSC gate). Then, the Ly6G^−^ population was examined for the expression of F4/80 macrophage-specific marker ((**A6**) Ly6G^−^ gate). Ly6C^+^/F4/80^−^ cells were thereafter tested for their MHCII expression (**A7**). The CD11b^+^/Ly6C^+^/F4/80^−^MHCII^−^ population was identified and sorted as Mo-MDSC ((**A7**) Mo-MDSC gate). (**B**) Percent of the cells identified as PMN-MDSCs (filled bar) and Mo-MDSCs (open bar) of the total CD11b^+^ cells in LLC tumors. Bars and error bars indicate mean ± SEM (PMN-MDSCs, *n* = 5; Mo-MDSCs, *n* = 5, and individual data points are indicated with circles). The ~30% of myeloid cells not accounted for include tumor-associated dendritic cells (TADC), macrophages (TAM), and MDSCs differentiating into TAMs.

**Figure 3 ijms-24-06216-f003:**
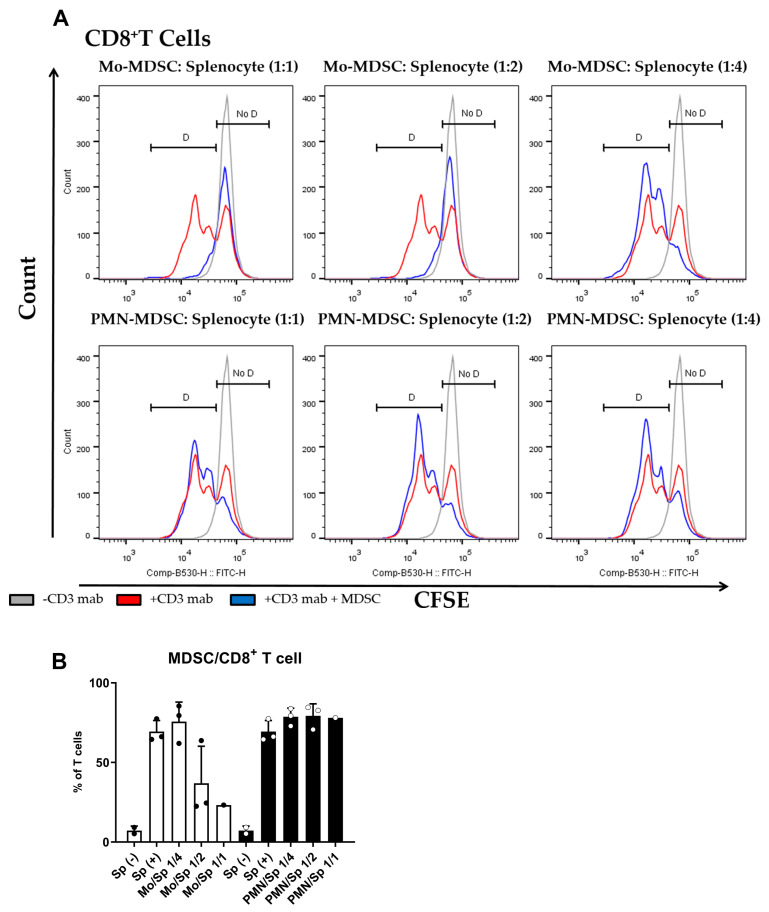
MDSC-mediated suppression of CD8^+^ T cell proliferation. (**A**) CFSE-labelled splenocytes were activated by anti-mouse CD3 antibody and co-cultured (blue) with Mo-MDSCs (upper row) or PMN-MDSCs (lower row) isolated freshly from LLC tumors at the indicated cell ratios. As controls, stimulated splenocytes (red) and unstimulated splenocytes (grey) were cultured separately. Generations of proliferating CD8^+^ T cells were identified using the histogram of unstimulated splenocytes as reference (D: proliferating cells, No D: non-proliferating cells). Histograms show one representative experiment out of three independent ones. (**B**) Quantification of the proliferation of CD8^+^ T cells in the presence of MDSCs. The percentage of proliferating CD8^+^ T cells in each condition was determined as D/(D + No D) × 100 where “D” and “no D” indicate the number of cells in the corresponding regions in panel A. PMN-MDSCs (empty circles and filled bars) and Mo-MDSCs (filled circles and empty bars) were used at indicated MDSC/splenocyte ratios and cultured following activation by anti-mouse CD3. Sp (+): anti-mouse CD3 stimulated splenocyte in the absence of MDSCs; Sp (−): unstimulated splenocytes in the absence of MDSCs. Bars and error bars indicate mean ± SD.

**Figure 4 ijms-24-06216-f004:**
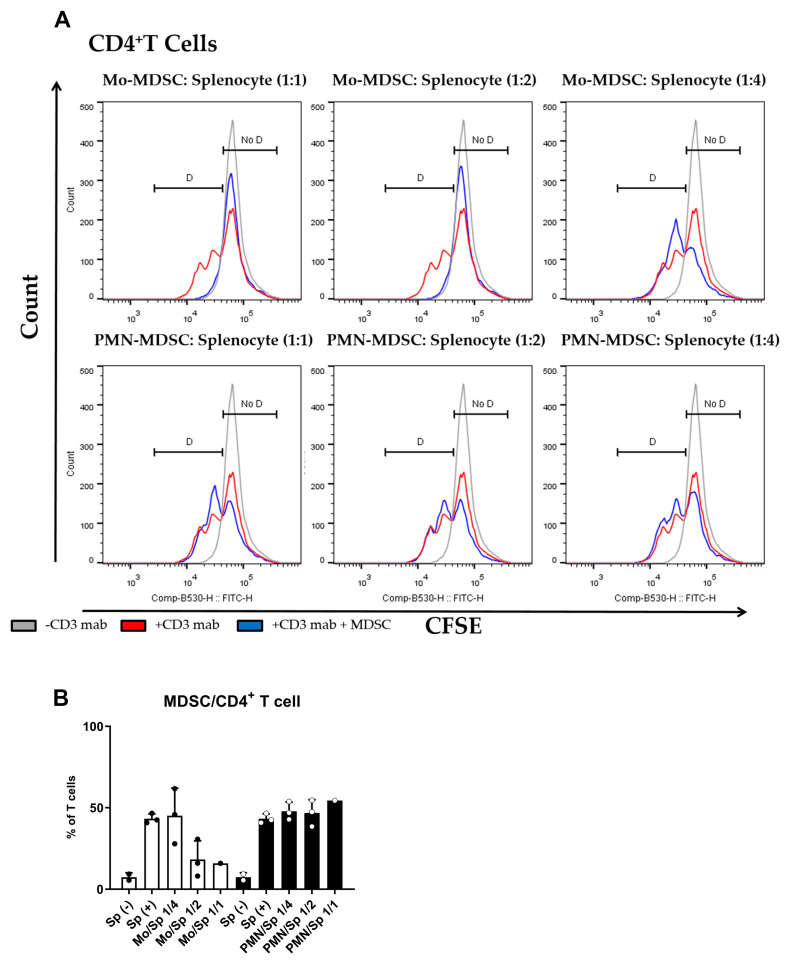
MDSC-mediated suppression of CD4^+^ T cell proliferation. (**A**) CFSE-labelled splenocytes were activated by anti-mouse CD3 antibody and co-cultured (blue) with Mo-MDSCs (upper row) or PMN-MDSCs (lower row) isolated freshly from LLC tumors at the indicated cell ratios. As controls, stimulated splenocytes (red) and unstimulated splenocytes (grey) were cultured separately. Generations of proliferating CD4^+^ T cells were identified using the histogram of unstimulated splenocytes as reference (D: proliferating cells, No D: non-proliferating cells). Histograms show one representative experiment out of three independent ones. (**B**) Quantification of the proliferation of CD4^+^ T cells in the presence of MDSCs. The percentage of proliferating CD4^+^ T cells in each condition was determined as D/(D + No D) × 100 where “D” and “no D” indicate the number of cells in the corresponding regions in panel A. PMN-MDSCs (empty circles and filled bars) and Mo-MDSCs (filled circles and empty bars) were used at indicated MDSC/splenocyte ratios and cultured following activation by anti-mouse CD3. Sp (+): anti-mouse CD3 stimulated splenocyte in the absence of MDSCs; Sp (−): unstimulated splenocytes in the absence of MDSCs. Bars and error bars indicate mean ± SD.

**Figure 5 ijms-24-06216-f005:**
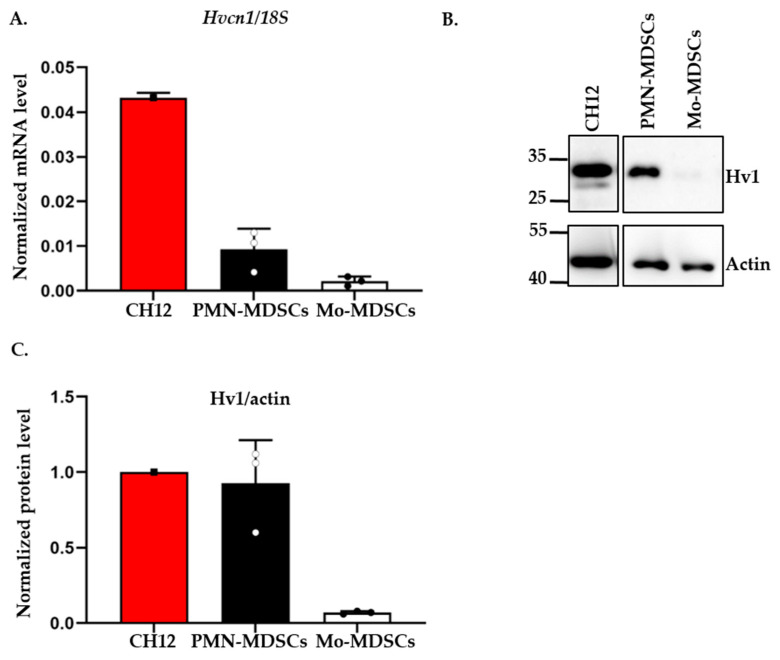
Expression of Hv1 proton channel by murine PMN-MDSCs and Mo-MDSCs at mRNA and protein level. (**A**) Hv1 RNA level relative to 18S RNA expressed by MDSCs using RT-qPCR. The individual data points (empty or filled circles) show the average of duplicates. RNA extracted from the CH12 B cell lymphoma cell line was used as a positive control. Bars and error bars indicate mean ± SD (PMN-MDSCs, *n =* 3; Mo-MDSCs, *n =* 3). (**B**) Western blot analysis of Hv1 proton channel protein from the sorted MDSCs: protein lysate from CH12 B cell lymphoma was used as a positive control and actin as a housekeeping protein. (**C**) Densiometric analysis of Hv1 protein level expressed by PMN- and Mo-MDSCs relative to actin expression. Densitometric data were obtained for 3 independent sets of sorted cells. Bars and error bars indicate mean ± SD (PMN-MDSCs, *n =* 3; Mo-MDSCs, *n =* 3).

**Figure 6 ijms-24-06216-f006:**
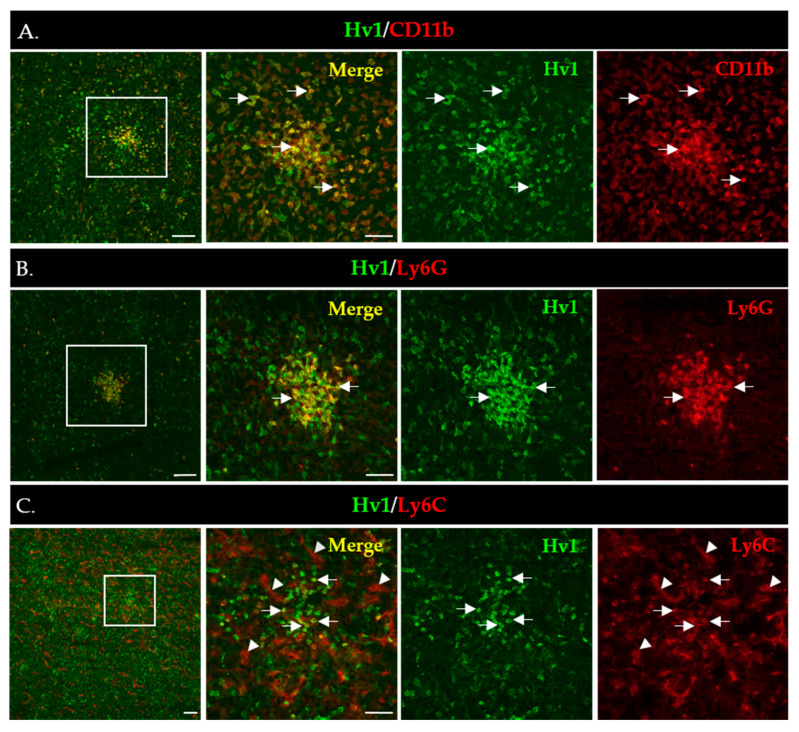
Immunofluorescence staining of LLC cryosections to detect expression of Hv1 by myeloid cells (**A**–**C**). Boxed areas in the 1st column are shown in higher magnification to the right. Scale bars, 100 µm for the lower and 50 µm for the higher magnification images. (**A**) CD11b (red) marks all myeloid cells accumulated in the LLC tumor (4th column); the Hv1 signal is in green (3rd column), whereas the CD11b/Hv1 co-expressing myeloid cells are in yellow in the merged images (2nd column). (**B**) Hv1^+^/Ly6G^+^ MDSCs (arrow) cells in an LLC tumor section. L6yG^+^ (red) marks PMN-MSDCs (4th column); the Hv1 signal is in green (3rd column), whereas the L6G^+^/Hv1 co-expressing myeloid cells are in yellow in the merged images (2nd column). (**C**) Hv1^+^/Ly6C^+^ cells (arrow) in a tumor section. Ly6C^+^ (red) marks Mo-MSDCs (4th column); the Hv1 signal is in green (3rd column), whereas the Ly6C^+^/Hv1 co-expressing myeloid cells are in yellow in the merged images (2nd column). Ly6C additionally marks the blood vessels (arrowheads).

**Figure 7 ijms-24-06216-f007:**
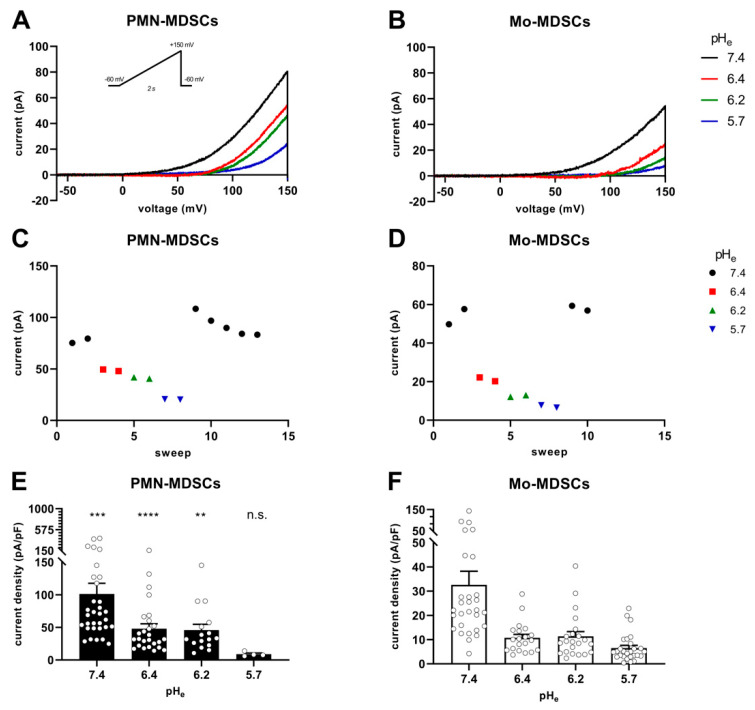
pH_e_-induced shift of Hv1 currents in murine PMN- and Mo-MDSCs. (**A**–**D**) Representative whole-cell current traces evoked by voltage ramps ranging from −60 to +150 mV, lasting 2 s and repeated every 15 s (see inset) in a PMN- (**A**) and a Mo-MDSC (**B**) freshly isolated from murine LLC. The currents at +145 mV were determined for every trace (see Materials and Methods) and plotted as a function of the sweep number. Currents obtained at different pH_e_ values for PMN- (**C**) and Mo-MDSCs (**D**) are represented by different colors and symbols (black circle: pH_e_ 7.4; red square: 6.4; green up triangle: 6.2; blue down triangle: 5.7). (**E**,**F**) Current density was calculated as the ratio of the current at +145 mV and the capacitance of the cell in pF determined from the readout of the amplifier whole-cell capacitance compensation circuit for each cell individually. Bars and error bars indicate the mean ± SEM (PMN-MDSCs–7.4: *n =* 32, 6.4: *n =* 26, 6.2: *n* = 16, 5.7: *n* = 4; Mo-MDSCs–7.4: *n =* 29, 6.4: *n =* 20, 6.2: *n* = 22, 5.7: *n* = 27) of the current densities of PMN- (**E**) and Mo-MDSCs (**F**) obtained at different pH_e_ values as indicated. Data between PMN- and Mo-MDSCs for every pH_e_ condition were compared via Student’s unpaired *t*-test with Welch’s correction and indicated in (**E**) (**, *p* < 0.01; ***, *p* < 0.001; ****, *p*< 0.0001).

**Figure 8 ijms-24-06216-f008:**
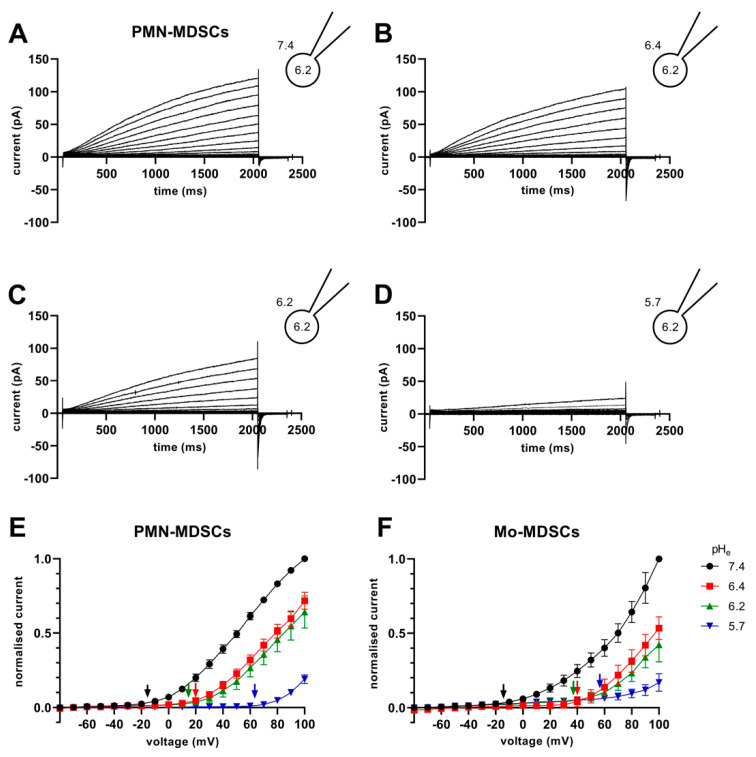
Families of H^+^ currents and I-V curves in murine PMN- and Mo-MDSCs. (**A**–**D**) Representative family of whole-cell H^+^ currents in murine PMN-MDSCs isolated freshly from LLC. Depolarizing 2 s long pulses were applied from a holding potential of −80 mV to +100 mV in 10 mV increments. While the internal solution was maintained constantly in the micropipette at pH_i_ 6.2, the cell was locally perfused at different pH_e_ (7.4 (**A**), 6.4 (**B**), 6.2 (**C**) and 5.7 (**D**)). Representative of 11 (**A**), 8 (**B**), 4 (**C**) and 3 (**D**) similar families of currents. Every trace has been filtered with a 25-point boxcar filter. (**E**,**F**) I–V curves for PMN- (**E**) and Mo-MDSCs (**F**) were built using the peak currents of every trace in every condition (black: pH_e_ 7.4; red: 6.4; green: 6.2; blue: 5.7) normalized to their respective current at +100 mV/pH_e_ = 7.4. The V_thr_ for every pH_e_ condition was calculated (see Materials and Methods, data analysis) and subsequently pointed out in the plot using an arrow with the associated color. Every value with its error bar indicates the mean ± SEM ((**E**)–7.4: *n* = 10–11, 6.4: *n =* 7–8, 6.2: *n* = 4, 5.7: *n* = 3. (**F**)–7.4: *n =* 5–9, 6.4: *n* = 4, 6.2: *n =* 2–4, 5.7: *n =* 4–5).

**Figure 9 ijms-24-06216-f009:**
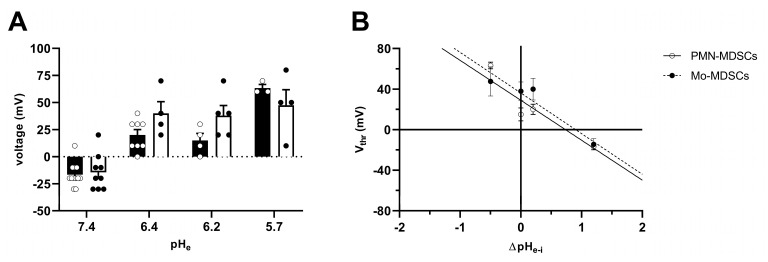
Effect of pH_e_ on the activation threshold (V_thr_) of Hv1 in murine PMN- and Mo-MDSCs. (**A**) Bars and error bars indicate the mean ± SEM (PMN-MDSCs–7.4: *n* = 12, 6.4: *n* = 8, 6.2: *n* = 4, 5.7: *n* = 3; Mo-MDSCs–7.4: *n* = 9, 6.4: *n* = 4, 6.2: *n* = 5, 5.7: *n* = 4) of the voltage threshold (V_thr_) calculated using the family of currents for each pH_e_ condition (see Materials and Methods and Figure 8E,F). The pH of the pipette filling solution was pH_i_ = 6.2. Symbols indicate individual data points obtained in PMN-MDSCs (empty circles and black-filled bars) and Mo-MDSCs (black-filled circles and empty bars). (**B**) V_thr_ plotted against ΔpH_e–i_ (pH_e_ − pH_i_) in PMN-MDSCs (empty circles) and Mo-MDSCs (black-filled circles). The shift in V_thr_ by moving pH_e_ of one unit was determined by the slope of the linear intercept (PMN-MDSCs: −39.36 mV/ΔpH, R^2^ = 0.7769; Mo-MDSCs: −40.07 mV/ΔpH, R^2^ = 0.6465). Every value with its error bar indicates the mean ± SEM.

**Figure 10 ijms-24-06216-f010:**
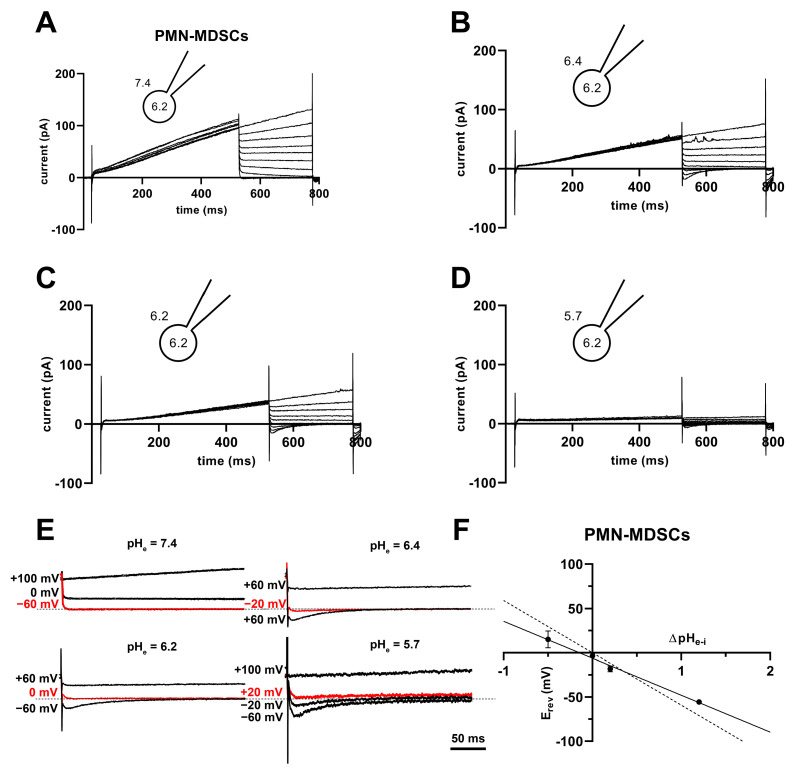
Tail currents and E_rev_ in PMN-MDSCs. (**A**–**D**) Representative families of currents obtained using the tail current protocol in a PMN-MDSC cell. The cell was depolarized from a holding potential of −80 mV to +100 mV for 500 ms followed by repolarizations to various potentials (from −60 to +100 mV in 20 mV increments). The pH of the internal solution was maintained pH_i_ = 6.2, and the cell was locally perfused with solutions having different pH_e_ of 7.4 (**A**), 6.4 (**B**), 6.2 (**C**) and 5.7 (**D**). (**E**) Selected tail currents from (**A**–**D**) are illustrated at increased time and amplitude resolution. The tail currents corresponding to E_rev_ are indicated in red. The dashed lines correspond to 0 pA current. (**F**) The reversal potentials (E_rev_) were determined individually for each cell at each ΔpH_e–i_, averaged (mean ± SEM, 7.4: *n =* 18, 6.4: *n =* 11, 6.2: *n =* 11, 5.7: *n* = 4), and plotted as a function of ΔpH_e–i_. The best fit linear regression (solid line, slope: −41.8 mV/ΔpH, R^2^ = 0.99) and the theoretical relationship calculated from the Nernst equation for H^+^ (dashed line, slope: −59.2 mV/ΔpH) are indicated.

**Figure 11 ijms-24-06216-f011:**
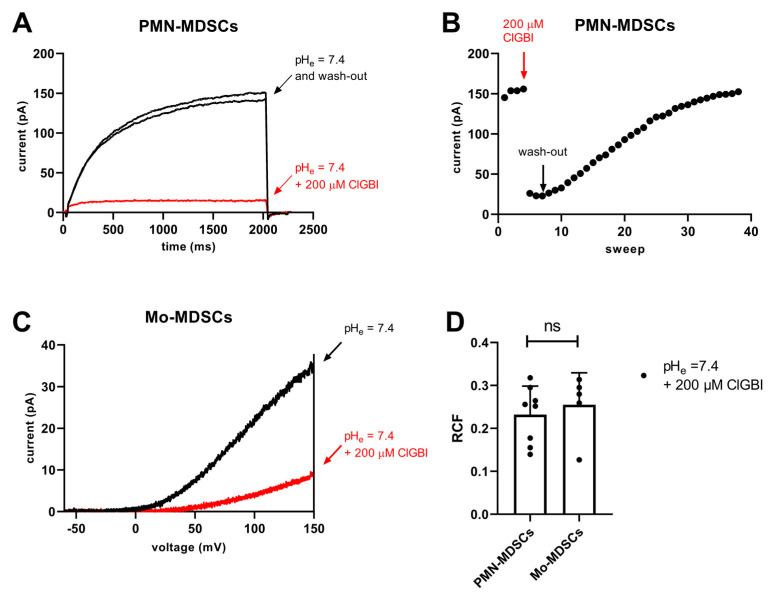
Inhibition of Hv1 currents in murine PMN- and Mo-MDSCs by ClGBI. (**A**) Representative whole-cell currents in a PMN-MDSs (holding potential: −80 mV, depolarization to +100 mV, every 15 s) prior to the application of ClGBI (black), at equilibrium block by 200 µM ClGBI (red) and following was-out (black). The pH_i_ and the pH_e_ were 6.2 and 7.4, respectively. (**B**) Representative time course of the onset and the recovery of the Hv1 current inhibition for the cell shown in panel A. Peak currents were determined as the average of the last 18 data points at +100 mV (see Materials and Methods); red and black arrows indicate the start of ClGBI application and the wash-out by ClGBI-free solution, respectively. (**C**) Representative whole-cell currents in a Mo-MDSCs (holding potential: −80 mV, voltage-ramp from −60 mV to +150 mV, every 10 s) prior to the application of ClGBI (black) and at equilibrium block by 200 µM ClGBI (red). Representative of 8 (**A**,**B**) and 5 (**C**) similar cells. Every trace has been filtered with a 25-point boxcar filter. (**D**) Remaining current fractions (RCF) in the presence of 200 μM ClGBI. Peak currents were determined for voltage-steps (PMN-MDSCs) and voltage-ramps (Mo-MDSCs) as described in the Materials and Methods. RCF was calculated as I/I_0_ where I_0_ and I are the peak currents in the absence and in the presence of 200 μM ClGBI upon reaching equilibrium block, respectively. Bars and error bars indicate the mean ± SEM, data points are determinations of RCF values in individual cells (PMN-MDSCs, *n* = 8; Mo-MDSCs, *n* = 5). Data were compared using Student’s *t*-test with Welch’s correction (*p* = 0.591, ns = not significant).

**Table 1 ijms-24-06216-t001:** List of antibodies used.

#	Antibody	Clone	Company	Cat. No.	Application
1	Rat anti mouse CD16/CD32	2.4G2	BD Pharmingen	553141	FC, FACS
2	Rat anti mouse CD11b-PE-Cy7	M1/70	eBioscience	25-0112-82	FACS
3	Rat anti CD11b-Alexa 594	M1/70	Biolegend	BZ 101254	IF
4	Rat anti CD11b-PercP.Cy5.5	M1/70	eBiosciece	101228	FC
5	Rat anti Ly6C-APC	HK1.1	Biolegend	128015	FACS, IF
6	Rat anti Ly6G-PE	1A8	BioLegend	127608	FACS
7	Rat anti Ly6G-APC	1A8	BioLegend		IF
8	Rat anti MHCII-PercP-Cy5	M5/114.15.2	BioLegend	107626	FACS
9	Rat anti F4/80 -AF488	CI:A3-1	BioRad	MCA497A488	FACS
10	Hamster anti mouse CD3e	145-2C11	Kind gift from Prof. Jo A. Van Ginderachter		T cell proliferation
11	Rat anti mouse CD4-PE	RM4-5	BD Pharmingen	553049	FC
12	Rat anti mouse CD8a-APC	clone53.6.7	eBioscience	17-0081-83	FC
13	Rabbit anti mouse Hv1	-	Alomone Labs	AHC-001	WB, IF
14	Mouse anti β-actin	C4	Santa Cruz Biotechnology	Sc-47778	WB
15	Sheep anti mouse IgG-HRP linked		GE Healthcare	NA931	WB
16	Donkey anti rabbit IgG (H&L) Alexa488 preadsorbed	-	Abcam	ab150061	IF
17	Donkey anti rabbit IgG-HRP linked		GE Healthcare	NA934	WB

## Data Availability

Research data are available upon request: panyi@med.unideb.hu.

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
