# Peer review of "The Voltage-Gated Hv1 H+ Channel Is Expressed in Tumor-Infiltrating Myeloid-Derived Suppressor Cells"

_ijms, 2023, doi:10.3390/ijms24076216_

Round 1

Reviewer 1 Report

The study by Cozzolino and colleagues  was performed in a group with long-lasting epxerience in the assessment of ion channel activity in diverse immune cells. This is a highly relevant paper that addressed for the first time the ion channels that are expressed in tumor-derived myeloid-derived suppressor cells that are currently in the focus in tumour biology as potential targets.

The experiments are performed in a very thorough way, with all controls in place, and the paper is written clearly and in a rigorous manner. The study employs different techniques, therefore it is important that the materials and methods section gives all necessary details in order to eventually replicate these experiments. In summary, the authors tackled an important biological question and reliably concluded that MDSC express Hv1 proton channels  with all biophysical characteristics.

I have only a few minor comments:

Please specify whether DNA-se was used when isolated RNA for the RT-PCR.

Why was phosphatase inhibitor coctail used in the samples loaded on Western blot?

Result section 3.1 The first sentence could be deleted, as this information is already given in the introduction section.

The authors write: "We determined the relative proportion of these two populations within the CD11b+ cells  in the tumor and detected a predominance of PMN-MDSCs (60%) compared to Mo-MDSCs (around 10%)". I was wondering what type of cells are the remnant cells (30%).

Figure 2. Please mention in the legend what cell types are those that  are not highlighted.

Are the 15 ms-long voltage steps lomg enough to detect all types of possibly expressed channels?

Line 693: The authors might want to mention what current level was detected in the two types of MDSC in the other study they mention.

Author Response

First of all, we would like to thank Reviewer 1 the time and efforts devoted to reviewing our manuscript. We are also grateful for the comments that we addressed point-by-point below:

  1. Please specify whether DNA-se was used when isolated RNA for the RT-PCR.

Yes, indeed, RNA samples were treated with RNA-se free DNA-se prior to cDNA synthesis as described by the manufacturer. A sentence with the specification of the DNA-se was inserted into the text.

The sentence in the revised version reads as: 

“Total RNA was isolated from the sorted cells using Trizol reagent (Life Technolo-gies/Thermo Fisher Scientific, Waltham, MA, USA), treated with RNAse-free DNase 1 (Thermofisher, AM2222) for 30 min at 37°C, followed by inactivation of the DNase at 75°C for 10 minutes. The DNase-treated RNAs were submitted to cDNA synthesis. cDNAs were synthesized and cDNA was synthetized using High-Capacity cDNA Reverse Transcrip-tion Kit (ThermoFischer Scientific, Waltham, MA, USA, cat #4368814) following the manufacturer´s instruction.”

  1. Why was phosphatase inhibitor coctail used in the samples loaded on Western blot?

Thank you for your comment. Although for the published experiments phosphatase inhibitors are not required, a fraction of the extracts were used for the detection of phosphoproteins in experiments not related to this project, in which phosphatase activities had to be blocked, and consequently we used them.

  1. Result section 3.1 The first sentence could be deleted, as this information is already given in the introduction section.

The Reviewer is right, we deleted the sentence and added some other relevant text to read as:

Increased number of MDSCs are correlated with bad/worse prognosis in multiple tumor types [3].

  1. The authors write: "We determined the relative proportion of these two populations within the CD11b+ cells  in the tumor and detected a predominance of PMN-MDSCs (60%) compared to Mo-MDSCs (around 10%)". I was wondering what type of cells are the remnant cells (30%).

Within the CD11b+ lineages, Ly6Chigh /MHCII-/ F4/80neg cells were identified as Mo-MDSCs. CD11b+/ Ly6Cmed/low/neg cells, including F4/80pos tumor-associated macrophages (TAM), and MDSCs differentiating into TAMs (“intermediate macrophages”). These populations together represent over 95 % of the CD11b+ population. (Figure 2. A6-7).

A sentence about the “missing” 30% of cells was included in the main text to read as: The remaining 30% of myeloid cells include tumor-associated dendritic cells (TADC), macrophages (TAM), and MDSCs differentiating into TAMs.

  1. Figure 2. Please mention in the legend what cell types are those that are not highlighted

We have included a sentence about the missing 30% of the cell in the figure legend as well. Thank you for the note.

  1. Are the 15 ms-long voltage steps long enough to detect all types of possibly expressed channels?

We agree with Reviewer 1 that 15 ms is rather a short depolarization. On the other hand, depolarization to +50 mV greatly speeds up the activation kinetics of the voltage-gated K+ currents and, as such, the chosen pulse-length and depolarization combination is sufficient to see most of the voltage-gated K+ currents expressed. Based on the activation kinetics of the currents these channels are definitively activated by the 15-ms pulses to +50 mV: Kv1.1, Kv1.2, Kv1.3, Kv1.4, Kv1.5, Kv1.6, Kv1.7, Kv1.8, Kv2.1, Kv2.2, Kv3.2, Kv3.3, Kv3.4, Kv4.1, Kv4.2, Kv4.3, Kv10.1 and the duration is marginally enough for the activation of Kv10.2. Using the 15-ms-long pulse, we may miss Kv7 currents when the channels are in complex with the auxiliary subunit. Furthermore, Supplementary figure S1 shows voltage ramps where the membrane was depolarized to extended periods. Each ramp duration was 2s, that would have allowed the recording of a slowly activating voltage gated K+ current. Unfortunately, we have not provided the ramp duration in the figure legend, we apologize for this. We have now included this information in the revised version. Therefore, we think that we have safely assumed the lack of voltage gated K+ currents in MDSCs.

  1. Line 693: The authors might want to mention what current level was detected in the two types of MDSC in the other study they mention.

Thank you very much for the note, however, MDSC subsets were not separately investigated by in the Alvear-Arias paper. The magnitude of the currents recorded by Alvear-Arias et al on the mixed population were comparable to the ones we recorded : from Fig. 2 in the Alvear-Arias paper we can see that with voltage steps from -90mV to  +130 mV they could get currents between ~200 pA and ~1 nA https://www.ncbi.nlm.nih.gov/pmc/articles/PMC9169626/). To address the comment of the Reviewer we have included this information in the revised version if the manuscript to read as:

Our electrophysiology results are consistent with the data described for in vitro produced MDSCs where Hv1 H+ currents of similar magnitude (between 200 pA and 1 nA at +130 mV) to our study were reported using patch-clamp in a mixed MDSC population [21].

We hope that Reviewer 1 will find our answers appropriate and the revised manuscript improved and suitable for publication in the revised form.

Reviewer 2 Report

The manuscript by Marco Cozzolino and colleagues describes an interesting study of the role of the voltage-gated proton channel Hv1 in myeloid-derived suppressor cells isolated from a murine lung tumor model. The study is well performed and well described. I have just a few minor comments:

A few acronyms are not written in full at first encounter. On page 6 the acronyms “MES” and “EGTA” are not written out.

On page 14 (bottom) and in the legend to Figure 11, the term pHo is used. Should it be pHe?

Page 15, line 453: Not necessary to repeat “Figure 8”.

Page 16, lines 476-7: “Vthr - ΔpHe−i is shifted to depolarized potentials for the currents recorded in Mo-MDSCs (Figure 9A).” I am not an expert in evaluating microcurrents and how pH differences can influence them, but I was able to read and understand the paper. However, I am not able to understand how Figure 9 supports this claim. It might be that other readers will also be challenged here.

On page 22, lines 665 -666, the sentence “Even if the selectivity of ClGBI towards Hv1 has not been assessed yet, it is widely used as an indicator of the presence of the Hv1 current various cell types” lacks an “in”.

In Figure 10E the fonts are too small and not readable.

Author Response

We would like to thank Reviewer 2 for taking time to review our manuscript. We are especially thankful for the comments, which were we address below:

  1. A few acronyms are not written in full at first encounter. On page 6 the acronyms “MES” and “EGTA” are not written out.

Thank you for the note, we have included the chemical names in the revised version of the manuscript.

MES (2-(N-morpholino)ethanesulfonic acid) (line 216), EGTA (ethylene glycol-bis(β-aminoethyl ether)-N,N,N′,N′-tetraacetic acid) (line 217)

  1. On page 14 (bottom) and in the legend to Figure 11, the term pHo is used. Should it be pHe?

Thank you very much for spotting our error, we have corrected this in the revised manuscript, moreover, we have found another instance of the same mistake. Thank you again.

  1. Page 15, line 453: Not necessary to repeat “Figure 8”.

Thank you very much, we have corrected the error.

  1. Page 16, lines 476-7: “Vthr - ΔpHe−i is shifted to depolarized potentials for the currents recorded in Mo-MDSCs (Figure 9A).” I am not an expert in evaluating microcurrents and how pH differences can influence them, but I was able to read and understand the paper. However, I am not able to understand how Figure 9 supports this claim. It might be that other readers will also be challenged here.

Thank you very much for your comment. We have extended the description of the figure as per the request of the Reviewer and added explanatory terms:

“This is indicated by a slight upward shift in the Vthr - ΔpHe-i relationship obtained for Mo-MDSCs (dashed line in Figure 9B) versus that for PMN-MDSCs (solid line in Figure 9B), i.e., at identical pH gradients the thresholds are more positive for Mo-MDSCs as compared to PMN-MDSCs.   “

  1. On page 22, lines 665 -666, the sentence “Even if the selectivity of ClGBI towards Hv1 has not been assessed yet, it is widely used as an indicator of the presence of the Hv1 current various cell types” lacks an “in”.

Thank you for the note, we have added the missing word.

  1. In Figure 10E the fonts are too small and not readable.

Thank you very much for the comment, we have redrawn the figure.

We hope that Reviewer 2 will find the revised manuscript improved and suitable for publication in the revised form.

Reviewer 3 Report

In this manuscript, the authors describe voltage-gated proton (Hv1 H+) channel in tumor-infiltrating myeloid-derived suppressor cells. They found that the tumor cells express Hv1 H+ channel at mRNA and protein levels. In addition, they directly showed that depolarization activated Hv1 H+ channel current, which were effectively inhibited by the inhibitor CIGBI. Overall, experimental results are solid and the manuscript is well written. Following points should be addressed for further consideration.

1. line 233, p7: Please explain about the leak current correction more precisely. A custom-built algorithm is not clear.

2. Figure7: IVs at 6.4 in Fig.7A and B have slight inward currents around +50 mV, suggesting the leak-subtraction induces the inward currents.

3. Figure S1: The presence of voltage-gated K+ and Na+ currents was assayed by 15 ms-long voltage steps (line 404). This 15 ms -long voltage jump is too short to activate time-dependent K+ current activation. In addition, the voltage-jump to +50mV is too positive to evaluate voltage-gated Na+ channel current correctly: the driving force of Na+ is smaller at +50 mV.

4. Figure 11 legend: It is not clear about application of paired Student's t-test with Welch's correction. Welch's correction is applied to unpaired data. In addition, * and ** are missing in Figure 11D.

Author Response

First of all, we would like to thank Reviewer 3 the careful reading of our manuscript. We are really grateful for the thoughtful comments, and addressed them below:

  1. line 233, p7: Please explain about the leak current correction more precisely. A custom-built algorithm is not clear.

Thank you very much for the comment. We have explained in the revised version the method in detail. We also reference our previous paper where the methods are detailed.

  1. Figure7: IVs at 6.4 in Fig.7A and B have slight inward currents around +50 mV, suggesting the leak-subtraction induces the inward currents.

We agree with the reviewer, indeed leak correction, even if done using statistical approach (i.e., linear regression to find leak conductance) can be imperfect, as it is seen in the red traces of Fig. 7A and B. On the other hand, this overcompensation seems to be small (a few pA) but seems to be more visible in Fig. 7B where the currents are very small. We do not think that this overcompensation of the leak will significantly influence our results, as the errors are in a few pA range.

  1. Figure S1: The presence of voltage-gated K+ and Na+ currents was assayed by 15 ms-long voltage steps (line 404). This 15 ms -long voltage jump is too short to activate time-dependent K+ current activation. In addition, the voltage-jump to +50mV is too positive to evaluate voltage-gated Na+ channel current correctly: the driving force of Na+ is smaller at +50 mV.

Thank you very much for the note. For the detection of the K+ currents we agree with Reviewer 3 that 15 ms is rather a short depolarization. On the other hand, depolarization to +50 mV greatly speeds up the activation kinetics of the voltage-gated K+ currents and, as such, the chosen pulse-length and depolarization combination is sufficient to see most of the voltage-gated K+ currents expressed. Based on the activation kinetics of the currents these channels are definitively activated by the 15-ms pulses to +50 mV: Kv1.1, Kv1.2, Kv1.3, Kv1.4, Kv1.5, Kv1.6, Kv1.7, Kv1.8, Kv2.1, Kv2.2, Kv3.2, Kv3.3, Kv3.4, Kv4.1, Kv4.2, Kv4.3, Kv10.1 and the duration is marginally enough for the activation of Kv10.2. Using the 15-ms-long pulse, we may miss Kv7 currents when the channels are in complex with the auxiliary subunit. Furthermore, Supplementary figure S1 shows voltage ramps where the membrane was depolarized to extended periods. Each ramp duration was 2s, that would have allowed the recording of a slowly activating voltage gated K+ current. Unfortunately, we have not provided the ramp duration in the figure legend, we apologize for this. We have now included this information in the revised version. Therefore, we think that we have safely assumed the lack of voltage gated K+ currents in MDSCs.

As for the detection of a Na+ current, the reviewer is right, there is basically 0 driving force at +50 mV. We were aware of it but unfortunately, we have included misleading information in the Materials and Methods. On the other hand, we have explained the correct approach in the Results section:  “Ion currents over a wider range of membrane potentials and depolarization durations were studied using voltage ramps. In these experiments (Figure S1) we did not see inward currents characteristic of the presence of voltage-gated Na+ or Ca2+ channels.”  If there was a voltage-gated Na+ current then we should have seen a downward deflection of the whole-cell ramp current recorded in Na+-containing solutions, especially around 0 mV where the larges Na+ current is usually seen using conventional voltage steps. This was not seen in any of our records. We agree that detection of voltage-gated Ca2+ current is more problematic as the ion concentrations and the charge carriers were not optimized for recording Ca2+ currents.

Based on the recommendation we have modified the Materials and Methods section and added explanatory sentences to the results to reflect some shortcomings of the study.

  1. Figure 11 legend: It is not clear about application of paired Student's t-test with Welch's correction. Welch's correction is applied to unpaired data. In addition, * and ** are missing in Figure 11D.

The Reviewer is right. This was a remnant of the text linked to Figure 7 and we have made and error by not correcting the statistical test after copying the relevant part of the legend. In this case we used unpaired Student’s t-test with Welch’s correction (ns, p = 0.591).  We have corrected the corresponding section in the manuscript and redrawn the graph.

We hope that Reviewer 3 will find the revised manuscript improved and suitable for publication in the revised form.